# *Trichoderma*-Bioenriched Vermicompost Induces Defense Response and Promotes Plant Growth in Thai Rice Variety “Chor Khing”

**DOI:** 10.3390/jof10080582

**Published:** 2024-08-16

**Authors:** Prisana Wonglom, On-Uma Ruangwong, Wasin Poncheewin, Siwaret Arikit, Kanamon Riangwong, Anurag Sunpapao

**Affiliations:** 1Faculty of Technology and Community Development, Thaksin University, Pa-Payom 93210, Thailand; prisana.w@tsu.ac.th; 2Department of Entomology and Plant Pathology, Faculty of Agriculture, Chiang Mai University, Mueang Chiang Mai 50200, Thailand; on-uma.r@cmu.ac.th; 3National Center for Genetic Engineering and Biotechnology (BIOTEC), National Science and Technology Development Agency (NSTDA), Khlong Luang 10120, Thailand; wasin.pon@biotec.or.th; 4Department of Agronomy, Faculty of Agriculture at Kamphaeng Saen, Kasetsart University, Kamphaeng Saen Campus, Kamphaeng Saen 73140, Thailand; arikit@gmail.com; 5Department of Biotechnology, Faculty of Engineering and Industrial Technology, Silpakorn University, Sanamchandra Palace Campus, Nakhon Pathom 73000, Thailand; 6Agricultural Innovation and Management Division (Pest Management), Faculty of Natural Resources, Prince of Songkla University, Hatyai, 90110, Thailand

**Keywords:** biocontrol agent, disease resistance, vermicompost, *Trichoderma asperelloides*, mycobiota

## Abstract

Vermicompost (VC) produced by African nightcrawler earthworms (*Eudrilus eugeniae*) is a natural fertilizer with a rich microbial community. *Trichoderma asperelloides* PSU-P1 is an effective antagonistic microorganism with multifaceted activity mechanisms. This research aimed to develop *Trichoderma*-bioenriched vermicompost (TBVC) to promote plant growth and induce the defense response in the Thai rice variety “Chor Khing”. *T. asperelloides* PSU-P1 was tested against *Rhizoctonia solani*, the pathogen of sheath blight disease, using a dual-culture assay. The results showed that *T. asperelloides* PSU-P1 effectively inhibited *R. solani* in vitro growth by 70.48%. The TBVC was prepared by adding a conidial suspension (10^8^ conidia/mL) to vermicompost. The viability of *Trichoderma* persisted in the vermicompost for 6 months and ranged from 1.2 to 2.8 × 10^7^ CFU/mL. Vermicompost water extracts significantly enhanced seed germination, root length, and shoot length compared to a control group (*p* < 0.05). Plants that received the TBVC displayed significantly longer shoot and root lengths and higher total chlorophyll content than control plants (*p* < 0.05). The TBVC induced defense response by increasing the enzyme activity of peroxidase (POD) and polyphenol oxidase (PPO) in comparison with control plants. Rice grown in the TBVC had a significantly reduced incidence of sheath blight caused by *R. solani* in comparison with control rice (*p* < 0.05). Furthermore, the fungal community of rice plants was analyzed via the high-throughput next-generation sequencing of the internal transcribed spacer (ITS). The fungal community in the TBVC had greater alpha diversity than the community in the VC. Phylum Ascomycota was dominant in both samples, and a heat map showed that *Trichoderma* was more prevalent in the TBVC than in the VC. Our results indicate that the enrichment of VC with *Trichoderma* increases growth, enhances the defense response, and reduces the incidence of sheath blight disease in the Thai rice variety “Chor Khing”.

## 1. Introduction

Vermicompost (VC) is a product of vermicomposting by earthworms, which convert agricultural waste material into compost as small rigid structures. This compost has been reported to enhance plant growth [1]. Earthworm compost is considered to be an organic fertilizer with high humus and nutrient contents and diverse soil microbes with the ability to enhance plant growth [2]. It contains 7.37% nitrogen and 19.85% phosphorus in the form of P_2_O_5_ [3]. The characteristics of vermicompost are able to stimulate microbial activity, improve soil quality, and increase the growth and yield of plants [4]. According to these abilities, vermicompost is widely used as a natural fertilizer to promote plant growth, as the uptake of its rich macro- and micro-nutrients has a strong impact on plant growth and development [5]. Vermicompost has been widely used to promote plant growth in several plant species. For instance, vermicompost enhances GA and IAA production, which significantly increases shoot length, the length of the internode, and the numbers of leaves and branches in *Capsicum annum* [6]. The application of vermicompost combined with vermiculite and sand increases the parameters of vegetative growth such as plant height, plant and root length, stem diameter, and fresh and dry weight in bananas [7]. The application of vermicompost and its derivatives has been utilized as a biological control method to control several plant diseases. For instance, vermiwash, which is a liquid extract from vermicompost, has been used to inhibit the mycelial growth of plant pathogenic fungi [8]. Furthermore, it has been found to prevent the development of powdery mildew in watermelons [9].

*Trichoderma* is a genus of filamentous fungi in the family Hypocreaceae that inhabits soil and plant tissues [10,11]. The genus *Trichoderma* comprises over 400 species, and some new species are listed by Species Fungorum. Recent discoveries have shown that *Trichoderma* species form symbiotic relationships with plants as endophytes [12,13] and act as pathogens in some plant species [14]. Several *Trichoderma* species have been used in agriculture to control plant diseases and promote plant growth. They have shown multifaceted mechanisms against plant pathogens such as competition for nutrients and space [15], antibiosis [16], inducing defense responses in plants [17], mycoparasites [18], producing cell wall-degrading enzymes [19], releasing volatile antifungal compounds [20], and increasing plant growth [21,22,23]. For instance, *T. longibrachiatum* and *T. asperelloides* have shown strong antagonistic activity against late wilt disease and increased the growth and yield of maize [24]. *Trichoderma harzianum* has been reported to enhance physiological and biochemical traits of cucumber seedlings, control Fusarium wilt disease, and increase the yield of cucumber fruits [25].

Rice is an economically important crop in Thailand, where there is a strong tradition for rice production. It is one of the main foods for consumption and a major agricultural export product [26]. In 2022, the production of all Thai rice was about 26.702 million tons with a trade value of approximately THB 109,363.2 million (Thai Rice Exports Association). The commercial rice cultivars mostly commonly exported to foreign countries are aromatic jasmine rice (i.e., Khao Dawk Mali 105 (KDML105)) and glutinous rice (i.e., RD6) [27]. Both rice cultivars are mainly cultivated in the northeast region of Thailand [28].

In southern Thailand, several local rice varieties are cultivated for consumption and sale, for instance, the “Chor Khing” variety, which has purple-red grains and high nutrient contents. This variety is mainly cultivated in Songkhla province in southern Thailand. The local rice in this area of southern Thailand is faced with slow growth and development. To support sustainable agriculture and achieve sustainable development goals (SDGs), the use of vermicompost in combination with a beneficial *Trichoderma* strain may be considered as an effective way to increase plant growth and induce defense responses against fungal pathogens. Therefore, this research aimed to promote plant growth and induce the defense response against *Rhizotonia solani*, the pathogen of sheath blight, in the rice variety “Chor Khing”.

## 2. Materials and Methods

### 2.1. Sources of “Chor Khing” Rice Variety and Preparation of Trichoderma-Bioenriched Vermicompost

Rice seeds of the “Chor Khing” cultivar were obtained from Phatthalung Rice Research Center (Rice Department, Ministry of Agriculture and Cooperatives). Vermicompost (VC) products from vermicomposting by African nightcrawler earthworms (*Eudrilus eugeniae*) were obtained from a social enterprise in Phatthalung province in southern Thailand. The worms were fed dairy cow manure and vegetables (cucumber and papaya). The effective *Trichoderma asperelloides* strain PSU-P1 [29] was obtained from the culture collection of the Pest Management Division, Faculty of Natural Resources, Prince of Songkla University, Thailand. *T. asperelloides* PSU-P1 was cultured on paddy rice in a glass bottle and incubated at ambient temperature (28 ± 2 °C) for 7 days to produce a mass of conidia. Then, 100 mL of sterilized distilled water (DW) was added to the bottle to harvest the conidia. The total concentration of *T. asperelloides* PSU-P1 was adjusted to 10^8^ conidia/mL with sterilized DW. Next, 100 mL of the spore suspension of *T. asperelloides* PSU-P1 was added to 1000 g of vermicompost to obtain *Trichoderma*-bioenriched vermicompost (TBVC), which was subjected to further bioassays.

### 2.2. Antifungal Activity of Trichoderma asperelloides PSU-P1 Against Rhizoctonia solani

It is known that *T*. *asperelloides* PSU-P1 effectively inhibits the mycelial growth of *Stagonosporopsis cucurbitacearum*, the pathogen of gummy stem blight in muskmelon [29]. *Rhizoctonia solani* was obtained from Department of Entomology and Plant Pathology, Faculty of Agriculture, Chiang Mai University. To test the antifungal ability of *T*. *asperelloides* PSU-P1 against *R. solani*, the pathogen of sheath blight in rice, a dual-culture assay was conducted on PDA plates. An agar plug from a 5-day-old colony of *R. solani* was placed directly on one side of a Petri dish containing PDA, and an agar plug of *T*. *asperelloides* PSU-P1 was placed on the opposite side at 5 cm away from *R. solani*. A Petri dish without an agar plug of *T*. *asperelloides* PSU-P1 served as a control. The dual-culture test had a completely randomized design (CRD) with three replications and was repeated twice. The tested Petri dishes were then incubated at ambient temperature for 7 days. The mycelial growth of *R. solani* was measured and converted to the percentage of fungal growth inhibition using the following formula:Growth inhibition (%) = [(R1 − R2)/R1] × 100
where R1 = colony radii of *R. solani* in the control, and R2 = colony radii of *R. solani* on the tested plate.

### 2.3. Viability of Trichoderma in Trichoderma-Bioenriched Vermicompost

To test the viability of the *Trichoderma* in the TBVC, this research was conducted for 6 months. A total of 1 g of TBVC was suspended in 100 mL of sterilized DW and diluted 10-fold. Then, 100 μL of the liquid phase was added to glucose ammonium nitrate agar (GANA) containing 1 g of glucose, 1 g of NH_4_NO_3_, 0.5 g of K_2_HPO_4_, 1 g of yeast extract, 0.5 g of Rose Bengal, and 15 g of DW. A total of 100 μL was placed directly on the GANA medium and subjected to the dilution spread plate technique. This experiment was conducted with 3 replicates. The Petri dishes were then incubated at ambient temperature for 72 h. *Trichoderma* colonies with fluffy greenish conidia on GANA media were counted each month for 6 months.

### 2.4. Effect of Trichoderma-Bioenriched Vermicompost Water Extracts on Seed Germination

A total of 10 g of TBVC was suspended in 10 mL of sterilized distilled water (DW) and centrifuged at 4000 revolutions per minute (rpm) at ambient temperature for 15 min. The supernatant was selected and filtered through Whatman filter paper (No. 1, 110 mm) to obtain TBVC water extract. A total of 22 rice seeds were placed directly on a Petri dish containing filter paper. Then, 2 mL of the crude supernatant from the formulation was applied to the filter paper in the Petri dish to maintain the humidity for seed germination. The application of sterilized DW served as a control. This experiment was conducted with three replicates. A Petri dish containing rice seeds was incubated at ambient temperature with natural light. After 2 and 3 days of incubation, the number of germinated seeds was recorded and converted to the percentage of germination, and the germinated rice samples were photographed. The lengths of the roots from the germinated seeds were measured in the rice treated with the TBVC water extracts in comparison with the control group.

### 2.5. Effect of Trichoderma-Bioenriched Vermicompost on Rice Seedling Growth

Rice seeds were surface-disinfected using 1% sodium hypochlorite (NaOCl) and washed with sterile DW to eliminate excess NaOCl. The planting materials comprised (i) vermiculites and peat moss at a ratio of 9:1, which served as a control, and (ii) vermiculites and TBVC at a ratio of 9:1. Five rice seeds were grown in each planting material in 5.5 × 6.5 × 7 cm pots. Each treatment comprised 5 replicates. The growth of the rice seedlings was measured after 10 days of cultivation. The parameters of rice seedling growth, including the shoot length, root length, and fresh weight, were measured. The total chlorophyll content was measured according to the method used by Palta et al. [30]. Rice shoots were cut into small pieces, soaked in 80% acetone, and incubated at 4 °C for 10 h. The liquid phase was measured with a UV-5300 UV/VIS spectrophotometer (METASH, Shanghai, China) at 663 and 645 nm, and the total chlorophyll content was calculated using the following formula:[(8.02 × A663) + (20.2 × 645)] × V/100 × W
where V is the volume of acetone, and W is the fresh weight of the shoot sample.

### 2.6. Inducing Defense Response in Rice Seedlings with Trichoderma-Bioenriched Vermicompost

In this study, we tested the defense response in the rice seedling stage by the measurement of defense-related enzyme activities. The rice seedlings from Section 2.6 were subjected to protein extraction. They were homogenized with a phosphate buffer at pH 6.0 for peroxidase (POD) activity or pH 7.5 for polyphenol oxidase (PPO) activity. The samples were then centrifuged at 14,000 rpm at 4 °C for 10 min, and the supernatants were selected. The peroxidase (POD) and polyphenol oxidase (PPO) activities were used to quantify the defense responses of the rice seedlings. The peroxidase activity was analyzed using 1% O-phenylenediamine (OPDA) as a substrate with 0.3% H_2_O_2_ [31]. Polyphenol oxidase was measured using catechol as a substrate [32]. The reaction mixtures were measured with a UV-5300 UV/VIS spectrophotometer (METASH, Shanghai, China) at 420 and 495 nm for POD and PPO activities, respectively. Enzyme activity was expressed as units (U)/mL.

### 2.7. Pot Experiment

The rice sheath blight pathogen, *R. solani*, was cultured on PDA and incubated at room temperature for 7 days to form sclerotia that were then used as an inoculum. Rice seeds of the “Chor Khing” variety were surface-disinfected in 5% sodium hypochlorite (NaClO) and washed with sterilized distilled water (DW) to remove excess NaOCl. The rice seeds were soaked in sterilized DW overnight to induce germination. They were then grown in vermiculite and peat moss, incubated at ambient temperature (28–32 °C) with natural light, and watered once a day. After 10 days of growth, the rice seedlings were transferred to a plastic pot (18 × 28 × 40 cm) containing sterilized silt–clay soil, incubated at ambient temperature with natural light, and watered once a day. After 14 days of incubation, the rice plants were subjected to an experiment using one seedling per pot. There were 4 treatments: (T1) soil alone, (T2) soil + TBVC (1:1), (T3) soil + sclerotia, and (T4) soil + TBVC (1:1) + *R. solani*. For inoculation, three sclerotia of *R. solani* were placed at the base of each rice seedling. This experiment was conducted with 5 replicates. Then, the tested rice plants were incubated at ambient temperatures with natural light and watered once a day for 30 days. The disease score was measured using the sheath blight incidence rating scale according to the method of the IRRI [33]. This score was converted to the percentage of disease incidence (PDI) using the following formula:$$\mathrm{PDI}=(\frac{Sum\;of\;individual\;rating}{No\;of\;leave\;examined\times Maximum\;disease\;scale})\times 100$$

### 2.8. DNA Extraction

Total genomic DNA was extracted from a TBVC sample and from VC as a control without plant roots. A total of 250 mg was ground using glass beads in 2 mL microcentrifuge tubes and extracted with a DNeasy ^®^ PowerSoil^®^ Pro Kit (Qiagen, Hilden, Germany) following the manufacturer’s instructions. The quality of the DNA was observed using 1% agarose gel electrophoresis. Total genomic DNA was subjected to immediate analysis or stored at −20 °C until further investigation.

### 2.9. PCR Amplification and High-Throughput Sequencing

The internal transcribed spacer (ITS) region of 18srRNA was amplified using ITS5-1737F and ITS2-2043, which were linked to Illumina adapters. PCR amplification was carried out according to the method used by Phoka et al. [11] with Phusion^®^ High-Fidelity PCR Master Mix (New England Biolabs, Ipswitch, MA, USA). PCR products showing distinct bands between 200 and 400 base pairs (bps) were selected for further analysis. They were then purified using a Qiagen Gel Extraction Kit (Qiagen, Inc., Valencia, CA, USA). Libraries were prepared using an NEBNext^®^ UltraTMDNA Library Prep Kit for Illumina (New England Biolabs, Ipswich, MA, USA) and quantified with Qubit Fluorometers (Thermo Fisher Scientific, Waltham, MA, USA), and Q-PCR was analyzed via Q, the Illumina platform (Illumina, Inc., San Diego, CA, USA).

### 2.10. Alpha Diversity Analysis

The complexity of the microbial diversity was analyzed according to the alpha diversity. The species richness, Shannon, Simpson, and InvSimpson indices were analyzed in QIIME (version 1.7.0) and demonstrated with R software (version 2.15.3).

### 2.11. Statistical Analysis

The results of mycelial growth, the viability of *Trichoderma* conidia, plant growth, and enzyme activities were subjected to a one-way analysis of variance (ANOVA). Statistically significant differences among the treatments were analyzed using Student’s *t*-test and Tukey’s test (*p* < 0.05).

## 3. Results

### 3.1. Antifungal Ability of Trichoderma asperelloides PSU-P1 against Rhizoctonia solani

A dual-culture assay was conducted to test the antifungal ability of *Trichoderma asperelloides* PSU-P1 against *R. solani*. *T. asperelloides* PSU-P1 effectively inhibited the mycelial growth of *R. solani* in the tested dual-culture plates. The radius of the mycelial growth was 2.06 ± 0.40 cm, which was significantly lower than that of the control (pathogen alone). The inhibition percentage of *T. asperelloides* PSU-P1 against *R. solani* was 70.48% (Figure 1).

### 3.2. Survival of Trichoderma in Trichoderma-Bioenriched Vermicompost

To assess the viability of all *Trichoderma* in the TBVC, the dilution spread plate technique was conducted on the GANA medium each month for 6 months, and the viability was expressed as colony-forming units (CFUs)/mL. The results show that the colonies of all *Trichoderma* on the GANA ranged from 1.2 to 2.8 × 10^7^ CFU/mL over the 6-month period. After one month, the number of *Trichoderma* conidia was observed to be 2.20 × 10^7^ CFU/mL, and this increased to the highest level after the second month (2.80 × 10^7^ CFU/mL). The number of *Trichoderma* conidia then decreased slightly in months 3–6, and the final amount after 6 months was 1.20 × 10^7^ CFU/mL (Figure 2).

### 3.3. Trichoderma-Bioenriched Vermicompost Enhances Seed Germination

To test the effect of TBVC water extract on the seed germination of the Thai rice variety “Chor Khing”, the percentages of germinated seeds and root lengths were measured (Figure 3). The results show that the percentage of the rice seed germination of plants that received TBVC extracts was 98.48% 2 days post-application (dpa), which was significantly higher than that of the control plants (84.84%). The rice seeds treated with TBVC had a percentage of shoot germination of 98.48% 3 dpa, whereas this value was only 46.96% in the control group. The root length was also measured 3 dpa, and the results show that the root length of the rice treated with TBVC water extract was 1.62 ± 0.05 cm, which was significantly longer than that of the control group at 1.27 ± 0.16 cm.

### 3.4. Trichoderma-Bioenriched Vermicompost Promotes Plant Growth in Rice Seedlings

The ability of vermicompost to promote plant growth in rice seedlings was measured via the shoot height, root length, and fresh weight of rice 10 days post-application (Figure 4). The application of TBVC significantly increased plant growth in rice seedlings compared to control plants (*p* < 0.05). The shoot height of the TBVC-treated rice was 18.82 ± 1.12 cm, which was significantly higher than that of the control rice (15.38 ± 0.91 cm); the root length of the TBVC-treated rice was 13.35 ± 1.67 cm, which was significantly longer than that of the control rice (10.52 ± 0.93 cm); and the fresh shoot weight of the TBVC-treated rice was 0.36 ± 0.03 g, which was significantly higher than that of the control rice (0.29 ± 0.01 g). The fresh root weight showed similar values of 0.21 ± 0.03 and 0.14 ± 0.05 g for the TBVC-treated and control rice, respectively. Furthermore, the TBVC-treated rice displayed a total chlorophyll content of 7.81 ± 0.01 mg, which was significantly higher than that of the control rice (4.77 ± 0.01 mg).

### 3.5. Defense-Related Enzyme Activities in Rice Seedlings

Assays of the defense-related enzymes POD and PPO were conducted to observe the defense responses in rice after the application of TBVC. The results show that in the shoots and roots, the POD activity of the rice cultivated in TBV was significantly higher than that of the control rice. In the shoots, the POD activities of the rice grown in TBVC and the control rice were 274.62 and 184.61 U/mL, respectively (Figure 5). In the roots, the POD activity was similar, with values of 675.85 and 620.10 U/mL (Figure 5). The PPO activity showed no statistical difference between the rice shoots grown in TBVC (7.83 U/mL) and the control rice (7.09 U/mL). However, the PPO activity of the rice grown in TBVC was significantly higher than that of the control rice. The PPO activities of the rice roots grown in TBVC and the control roots were 10.04 and 1.32 U/mL, respectively (Figure 5).

### 3.6. Effect of Trichoderma-Bioenriched Vermicompost on Biological Control

To assess the effect of *Trichoderma*-bioenriched vermicompost in controlling sheath blight disease caused by *R. solani*, a pot experiment with four treatments was conducted. After 30 days of incubation, treatment T2 (soil + pathogen) had the highest disease incidence of 82.22%, whereas treatment T4 (soil + TBVC + pathogen) demonstrated a significantly lower PDI of 46.67%. However, without pathogen inoculation, treatments T1 (soil alone) and T3 (soil + TBVC) had incidences of 0% (Figure 6).

### 3.7. Fungal Community in Trichoderma-Bioenriched Vermicompost

The highest number of soil fungus sequences was obtained from the TBVC, which had more sequences than the VC (Figure 7). The *Trichoderma*-bioenriched vermicompost had the highest community richness, followed by the VC, with species richness numbers of 264 and 238, respectively (Figure 7). The Shannon, Simpson, and InvSimpson community indices indicated higher fungal community diversity in the TBVC than in the VC (Figure 7). The Shannon, Simpson, and InvSimpson values were 3.15, 0.90, and 10.70, respectively, in the TBVC, whereas values of 2.46, 0.82, and 5.68, respectively, were observed in the control sample.

In this study, only sequences belonging to the kingdom fungi were analyzed, and fungal operational taxonomic units (OTUs) were assigned to six different phyla, including Ascomycota, Basidiomycota, Chtridiomycota, Mortierellomycota, Mucoromycota, and Rozellomycota (Figure 8). Although six genera were observed in both samples, their relative abundances differed between the samples. In the VC, the top three phyla, Ascomycota, Basidiomycota, and Mortierellomycota, had relative abundances of 94.08%, 1.66%, and 0.17%, respectively. In the TBVC, the top three phyla, Ascomycota, Mucoromycota, and Basidiomycata, had relative abundances of 89.22%, 5.05%, and 1.71%, respectively (Figure 8). The distributions of the fungal genera in each sample are shown in Figure 9. The most abundant genera were different in the samples, and they are represented by dark-brown squares in the heat map. *Remersonia* was the most abundant genus in both samples, but its abundance was higher in the VC than in the TBVC. In contrast, the genus *Trichoderma* was found to be more abundant in the TBVC than in the VC (Figure 9).

## 4. Discussion

In this study, we developed *Trichoderma*-bioenriched vermicompost (TBVC) by combining the effective biocontrol agent *T. asperelloides* PSU-P1 and vermicompost in order to induce the defense response and promote plant growth in the rice variety “Chor Khing”. *T*. *asperelloides* PSU-P1 effectively suppressed the growth of *R. solani*, the pathogen of rice sheath blight. The viability of *Trichoderma* was measured for six months, and our results showed the survival of *Trichoderma* conidia until the sixth month. The application of TBVC induced the defense response in rice seedlings via the elevated enzyme activities of POD and PPO. The TBVC promoted plant growth in “Chor Khing” rice seedlings by increasing rice germination, shoot and root lengths, the fresh weights of roots and shoots, and the chlorophyll content. Furthermore, high fungal community diversity was observed in the TBVC.

The biocontrol agent *T*. *asperelloides* PSU-P1 has been reported to enhance the defense response in muskmelon plants against gummy stem blight [29] and was developed as a ready-to-use emulsion formulation [34] with viability against dragon fruit stem cankers for up to 6 months. In this study, *T*. *asperelloides* PSU-P1 effectively inhibited *R. solani.* Competition and mycoparasitic mechanisms against *R. solani* were revealed in an in vitro test. This result suggests that *T. asperelloides* PSU-P1 not only induces defense responses in plants [29] but is also able to compete for nutrients and space against phytopathogenic fungi, as observed in this study. However, according to the native *Trichoderma* in the vermicompost, we counted the viability of all *Trichoderma* in the TBVC. The *Trichoderma* in the TBVC had the highest CFU value in the second month, and this value gradually decreased until the sixth month. This may have occurred because the conidia of *Trichoderma* localized and reproduced in the TBVC in the first two months (log phase). After that, their number declined gradually according to the limitations of the nutrient source in the TBVC. However, a *Trichoderma* conidia concentration of 1.20 × 10^7^ CFU/mL was observed in the sixth month, which may be sufficient to suppress a fungal pathogen [34].

Using vermicompost as a natural fertilizer has also been reported to enhance growth and development in several plant species. Water-extractable fractions obtained from vermicompost have been shown to reduce the numbers of nematodes that cause root knot in tomatoes and bell peppers [35]. Using vermicompost water extract induced seed germination and seedling growth in tomatoes and lettuce [36]. Our results also showed that treatment with the water-extractable fraction of TBVC increased seed germination and root length in the Thai rice variety “Chor Khing”. Furthermore, vermicompost has been reported to increase commercial yields, total shoot biomass, and root biomass [37]. Our results are in agreement with these reports. We found that treatment with TBVC increased rice seedling growth based on the shoot and root lengths, the fresh weights of shoots and roots, and the total chlorophyll content. These results suggest that TBVC water extract induces seed germination and that the application of vermicompost increases the growth of “Chor Khing” rice seedlings.

Defense responses are induced in plants exposed to abiotic and biotic stresses [38]. Peroxidase (POD) and polyphenol oxidase (PPO) are defense-related enzymes involved in the plant defense process. POD acts in the early plant infection stage [39] and affects lignification and wound healing by promoting cell wall strengthening, which makes it difficult for pathogens to invade plants. PPO generates physical barriers around injured tissues by producing accumulations of melanin via polymerization [40,41]. Different parts of the same plant have different enzyme activities. For instance, more PPO activity against downy mildew disease was found in the shoots and leaves of pearl millet than in the roots [42]. POD activity was found to be higher in shoots than in roots during the seedling growth of barley, sweet peas, maize, and sorghum [43]. Our results showed that the activities of both enzymes were different in the roots and shoots. We compared the activities of both enzymes in the control samples and after treatment with TBVC. Our results showed that the POD activity in the treated rice was significantly higher than in the control rice in both the shoots and roots. On the other hand, the PPO activities in the roots were similar. This result revealed that application of TBVC induced defense-related enzyme activity (POD and PPO) in the plants. High enzyme activity and the accumulation of defense-related enzymes may facilitate resistance against fungal pathogens.

The vermicompost contained high fungal community diversity, as observed via the high-throughput sequencing of the ITS region. For instance, the fungal community of the vermicompost was almost exclusively dominated by the phylum Mortierellomycota, followed by Basidiomycota and Ascomycata [44]. At the genus level, *Mortierella* had the highest relative abundance in the vermicompost [44]. Our results show that the phylum Ascomycota had the highest relative abundance, followed by Mucoromycota and Basidiomycata. The phylum Ascomycota is not only dominant in vermicompost but also in soil samples observed in the southern part of Thailand [11]. The most abundant genus observed in this study was *Remersonia*. However, different feeding materials and different strains of earthworms result in different fungal communities in vermicompost samples [45]. Our results indicate that the genus *Trichoderma* had low levels of relative abundance in both samples. However, the techniques amplified the DNA in the samples randomly, so competitive amplification may have resulted in the low relative abundances of the genus *Trichoderma* in both samples. However, the TBVC had a slightly higher relative abundance of the genus *Trichoderma* than the VC, as indicated by the heat map analysis. Therefore, enriching vermicompost with *Trichoderma* conidia may result in the long viability and higher relative abundance of this genus.

Vermicompost has been applied to promote plant growth and suppress plant diseases in various crops. For instance, the application of vermicompost increased the plant biomass of *Dracocephalum moldavica* in peat-amended vermicompost (448%) and soil amended with vermicompost (68%) [46]. A vermicompost water extract effectively inhibited *R. solani* and reduced the incidence of damping-off in cucumber seedlings [47]. Vermicompost in combination with *T. asperellum* T13 effectively controlled sclerotium rot disease on soybean seedlings [48]. Our results are in agreement with these previous studies, as *Trichoderma*-bioenriched vermicompost (TBVC) induced disease resistance against sheath blight disease in the Thai rice variety “Chor Khing” and reduced the disease incidence.

## 5. Conclusions

In this study, we developed *Trichoderma*-bioenriched vermicompost in order to support rice growth and induce the defense response against *R. solani*, the pathogen that causes sheath blight in rice. The antagonist *T. asperelloides* PSU-P1 revealed antifungal activity against *R. solani*. The application of *Trichoderma*-bioenriched vermicompost enhanced seed germination, increased rice seedling growth, induced defense-related enzymes, and reduced the incidence of sheath blight. The present study suggests that *Trichoderma*-bioenriched vermicompost has the potential to be used as a natural fertilizer to prevent plant disease and promote plant growth. However, application in the field and comparisons with chemical fertilizers and synthetic fungicides need to be carried out in the near future.

## Figures and Tables

**Figure 1 jof-10-00582-f001:**
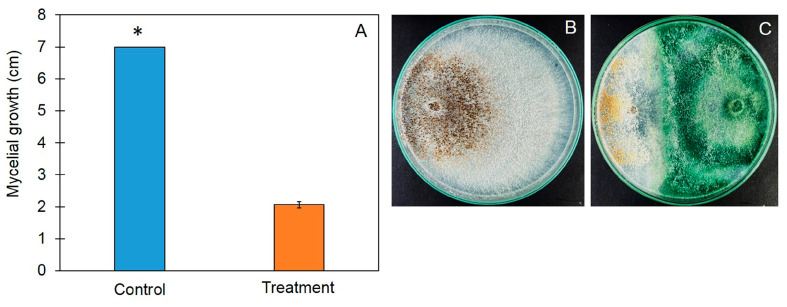
Mycelial growth of *Rhizoctonia solani* (**A**) and culture of *Trichoderma asperelloides* PSU-P1 and *R. solani* in control (**B**) and dual-culture (**C**) plates. Asterisk indicates significant differences between control and treatment groups according to Student’s *t*-test (*p* < 0.05).

**Figure 2 jof-10-00582-f002:**
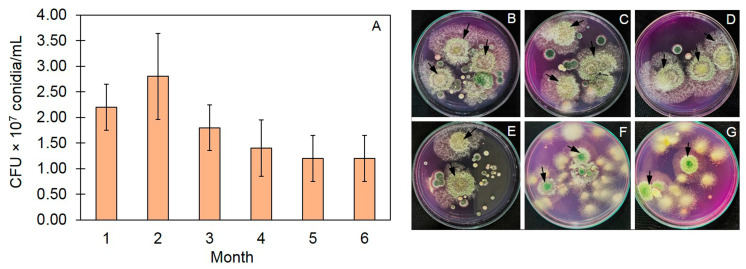
Viability of *Trichoderma* conidia in *Trichoderma*-bioenriched vermicompost (expressed as colony-forming units) (**A**) and colonies of *Trichoderma* on GANA medium after 1 to 6 months (**B**–**G**). Arrows indicate *Trichoderma* colonies on GANA medium.

**Figure 3 jof-10-00582-f003:**
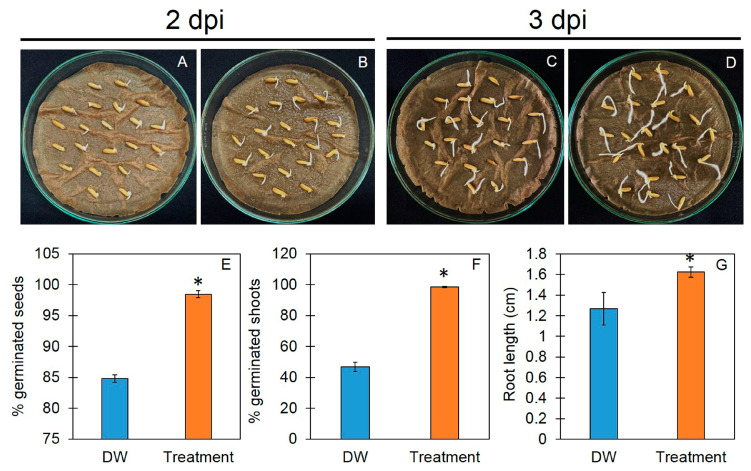
Effect of vermicompost water extract on seed germination of Thai rice. Seeds germinated in distilled water (**A**,**C**) and vermicompost water extract (**B**,**D**) at 2 and 3 days post-incubation (dpi). Quantitative analysis of germinated seeds (**E**), germinated shoots (**F**), and root lengths (**G**). Values are expressed as means ± SDs, and asterisks indicate significant differences between control and treatment groups according to Student’s *t*-test (*p* < 0.05).

**Figure 4 jof-10-00582-f004:**
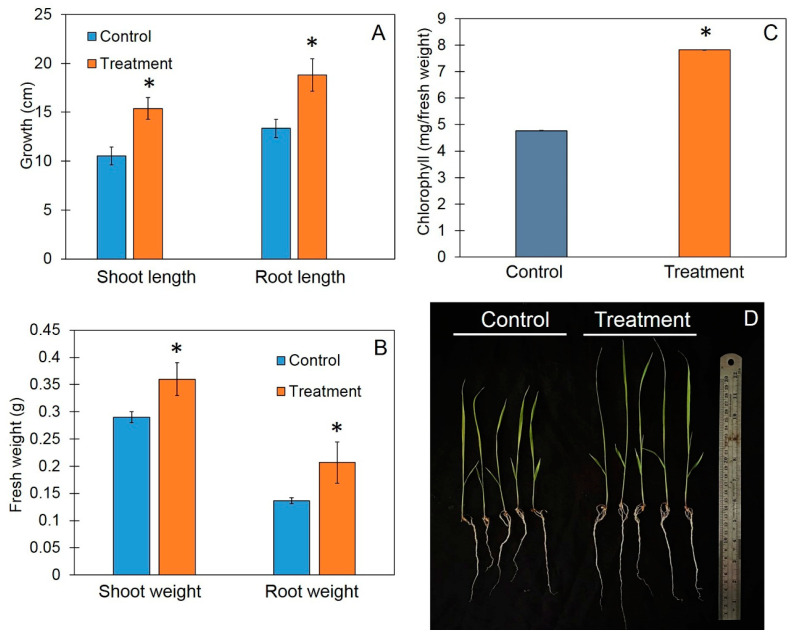
Effects of *Trichoderma*-bioenriched vermicompost on growth (**A**), fresh weight (**B**), chlorophyll content (**C**), and phenotype of seedlings (**D**). Values are expressed as means ± SDs, and asterisks indicate significant differences between control and treatment groups according to Student’s *t*-test (*p* < 0.05).

**Figure 5 jof-10-00582-f005:**
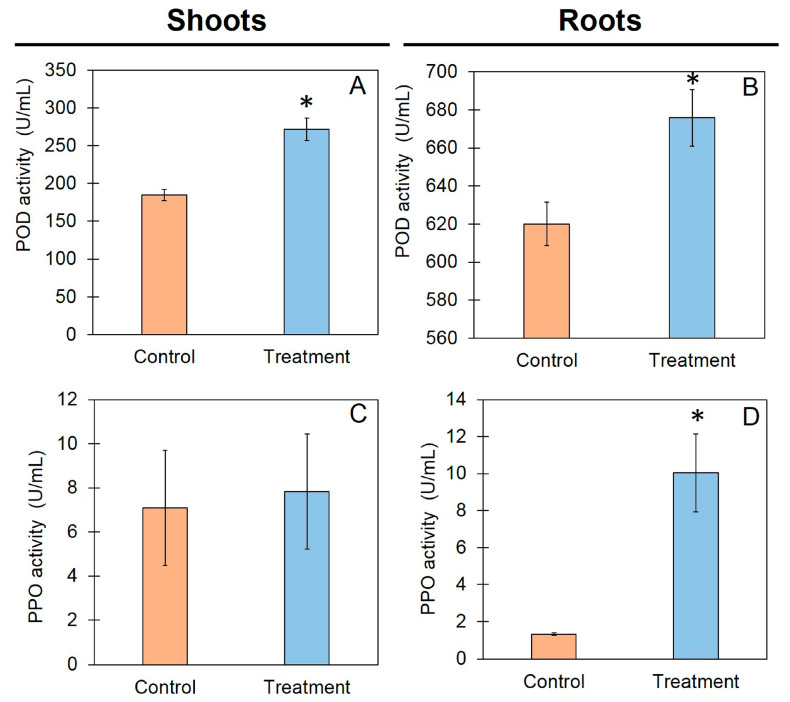
Defense-related enzyme activities in shoots and roots of rice variety “Chor Khing” grown in vermiculite (control) and vermiculite + Trichoderma-bioenriched vermicompost (treatment): peroxidase (POD) activities in shoots (**A**) and roots (**B**) and polyphenol oxidase (PPO) activities in shoots (**C**) and roots (**D**). Values are expressed as means ± SDs, and asterisks indicate significant differences between control and treatment groups according to Student’s *t*-test (*p* < 0.05).

**Figure 6 jof-10-00582-f006:**
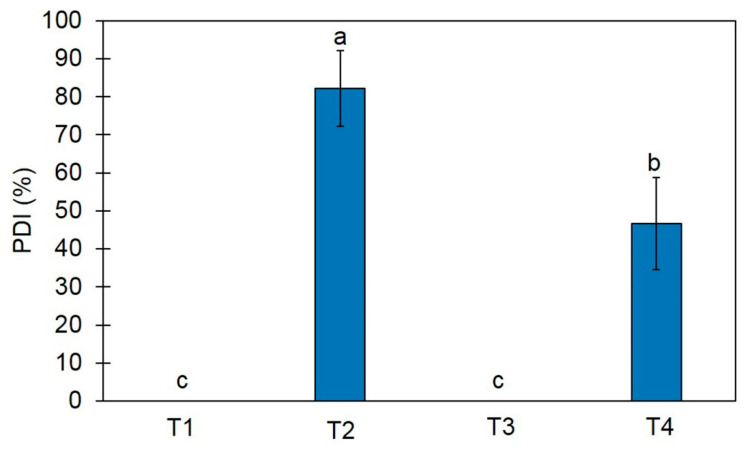
Percentages of disease incidence (PDIs) of rice variety “Chor Khing” grown in soil (T1), soil + sclerotia (T2), soil + TBVC (T3), and soil + TBVC + sclerotia (T4). Values are expressed as means ± SDs, and letters indicate significant differences between control (DW) and treatment groups according to Tukey’s test (*p* < 0.05).

**Figure 7 jof-10-00582-f007:**
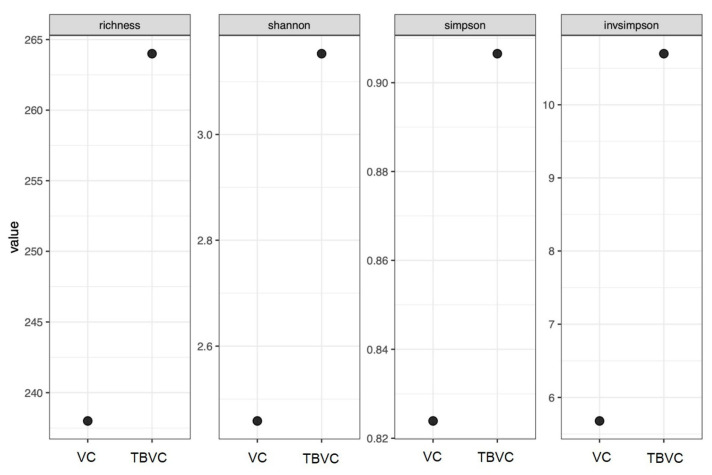
Species richness and alpha diversity indices of fungal communities in vermicompost (VC) and *Trichoderma*-bioenriched vermicompost (TBVC).

**Figure 8 jof-10-00582-f008:**
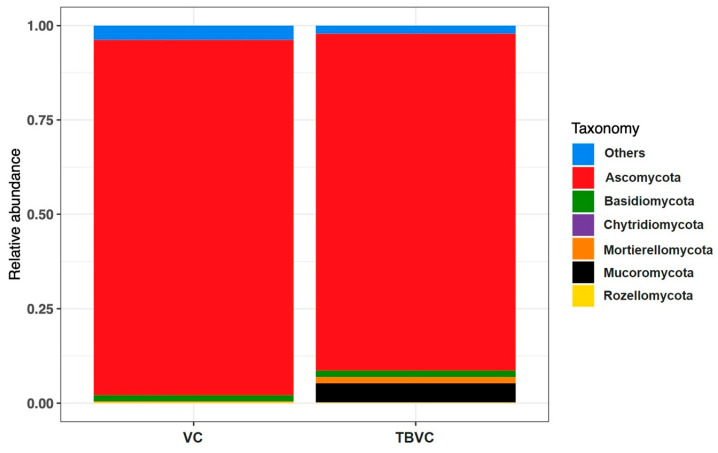
The relative abundances of dominant fungal phyla observed in the control sample and vermicompost. The relative abundances are based on the proportion frequencies of ITS DNA sequences.

**Figure 9 jof-10-00582-f009:**
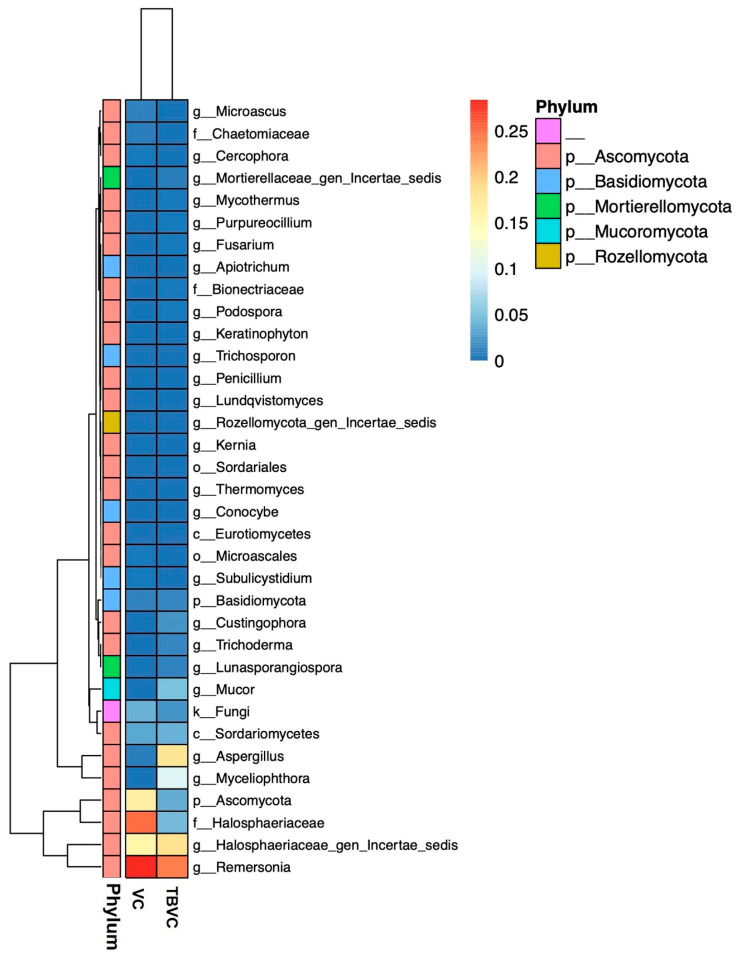
A heat map of the genus distributions in the control vermicompost (VC) and *Trichoderma*-bioenriched vermicompost (TBVC) samples. The dendrogram indicates a lack of similarly in the sequence region.

## Data Availability

The original contributions presented in the study are included in the article, further inquiries can be directed to the corresponding author.

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
