# Peer review of "Trichoderma-Bioenriched Vermicompost Induces Defense Response and Promotes Plant Growth in Thai Rice Variety “Chor Khing”"

_jof, 2024, doi:10.3390/jof10080582_

Round 1

Reviewer 1 Report

Manuscript entitled: Trichoderma-Bioenriched Vermicompost Induces Defense Re-2 sponse and Promotes Plant Growth in Thai Rice Variety “Chor 3 Khing”. The manuscript developed a Trichoderma-bioenriched vermicompost (TBVC) to promote plant growth and induce defense response in the Thai rice variety “Chor Khing”. Several points need to be addressed before it can be accepted.

1. Line 42. Replaced “natural compost” to “Vermicompost”.

2. Line 46-60. The application of VC on plant growth promotion had been introduced. The author should also introduce the reports of VC on biocontrol diseases in this section.

3. Line 133. The source of Rhizoctonia solani should be introduced.

4. Line 181. NaClO.

5. Line 188. Why didn’t set treatments of soil+VC, or soil+VC+ R. solani? 

6. Line 190. The author used clear why use sclerotia as inoculum, but not use mycelia? Sclerotia of R. solani firstly germinate to mycelia and then infect rice plants. 

7. Section 2.9 should move to behind of 2.10. 

8. Line 198. What is the meaning for comparison of the diversity between VC and TBVC? As the difference between VC and TBVC is T. asperelloides added in TBVC group. Obviously the most abundance species in TBVC group is T. asperelloides. And the results showed that difference exist on the community consist between VC and TBVC. Are the difference were due to T. asperelloides addition? Is T. asperelloides had effects on the community consist?

9. Some values should add ± SDs, such as Line 229: 2.06 cm, as well as Line 255, 256, 267, 268, 269, 270, 271, 272, 273, 274. 

10. Figure 2. Trichoderma should be labeled in each figure (B-G).

11. Line 310. Commonly the higher Shannon value and lower Simpson value represent higher alpha diversity.

Manuscript entitled: Trichoderma-Bioenriched Vermicompost Induces Defense Re-2 sponse and Promotes Plant Growth in Thai Rice Variety “Chor 3 Khing”. The manuscript developed a Trichoderma-bioenriched vermicompost (TBVC) to promote plant growth and induce defense response in the Thai rice variety “Chor Khing”. Several points need to be addressed before it can be accepted.

1. Line 42. Replaced “natural compost” to “Vermicompost”.

2. Line 46-60. The application of VC on plant growth promotion had been introduced. The author should also introduce the reports of VC on biocontrol diseases in this section.

3. Line 133. The source of Rhizoctonia solani should be introduced.

4. Line 181. NaClO.

5. Line 188. Why didn’t set treatments of soil+VC, or soil+VC+ R. solani? 

6. Line 190. The author used clear why use sclerotia as inoculum, but not use mycelia? Sclerotia of R. solani firstly germinate to mycelia and then infect rice plants. 

7. Section 2.9 should move to behind of 2.10. 

8. Line 198. What is the meaning for comparison of the diversity between VC and TBVC? As the difference between VC and TBVC is T. asperelloides added in TBVC group. Obviously the most abundance species in TBVC group is T. asperelloides. And the results showed that difference exist on the community consist between VC and TBVC. Are the difference were due to T. asperelloides addition? Is T. asperelloides had effects on the community consist?

9. Some values should add ± SDs, such as Line 229: 2.06 cm, as well as Line 255, 256, 267, 268, 269, 270, 271, 272, 273, 274. 

10. Figure 2. Trichoderma should be labeled in each figure (B-G).

11. Line 310. Commonly the higher Shannon value and lower Simpson value represent higher alpha diversity.

Author Response

Manuscript entitled: Trichoderma-Bioenriched Vermicompost Induces Defense Response and Promotes Plant Growth in Thai Rice Variety “Chor Khing”. The manuscript developed a Trichoderma-bioenriched vermicompost (TBVC) to promote plant growth and induce defense response in the Thai rice variety “Chor Khing”. Several points need to be addressed before it can be accepted.

Response: Thank you for reviewing this manuscript and providing valuable comments to improve it.

  1. Line 42. Replaced “natural compost” to “Vermicompost”.

Response 1: We have replaced as vermicomspost.

  1. Line 46-60. The application of VC on plant growth promotion had been introduced. The author should also introduce the reports of VC on biocontrol diseases in this section.

Response 2: We have added “The application of vermicompost and its derivatives has been utilized as a biological control method to control several plant diseases. For instance, vermiwash, which is a liquid extract from vermicompost, has been used to inhibit the mycelial growth of plant pathogenic fungi (Gudeta et al., 2021). Furthermore, it has been found to prevent the development of powdery mildew in watermelons (Naidu et al., 2012)”.

  1. Line 133. The source of Rhizoctonia solanishould be introduced.

Response 3: We have added “Rhizoctonia solani was obtained from Department of Entomology and Plant Pathology, Faculty of Agriculture, Chiang Mai University.”

  1. Line 181. NaClO.

Response 4: We have revised as “NaClO”.

  1. Line 188. Why didn’t set treatments of soil+VC, or soil+VC+ R. solani? 

Response 5: We have revised as suggestion “T1) soil alone, T2) soil + TBVC (1:1), T3) soil + sclerotia, and T4) soil + TBVC (1:1) + R. solani.”

  1. Line 190. The author used clear why use sclerotia as inoculum, but not use mycelia? Sclerotia of R. solani firstly germinate to mycelia and then infect rice plants. 

Response 6: As Rhizoctonia solani develops disease on plants through three forms of inoculum: basidiospores, mycelium fragments, and sclerotia. On this study we used sclerotia due to we can count number of sclerotia for inoculum. For mycelia inoculum it contain PDA of agar plug, which may easy to contaminate with other fungi.

  1. Section 2.9 should move to behind of 2.10. 

Response 7: We have revised as suggested.

  1. Line 198. What is the meaning for comparison of the diversity between VC and TBVC? As the difference between VC and TBVC is T. asperelloides added in TBVC group. Obviously the most abundance species in TBVC group is T. asperelloides. And the results showed that difference exist on the community consist between VC and TBVC. Are the difference were due to T. asperelloidesaddition? Is T. asperelloides had effects on the community consist?

Response 8: In this study, we aimed to investigate if the addition of T. asperelloides to vermicompost can alter the community structure of soil fungi. To investigate this hypothesis, we compared the relative abundance of soil fungi in vermicompost samples between VC (vermicompost alone) and TBVC (vermicompost with T. asperelloides). The results of the study indicated that the TBVC sample showed a higher abundance of certain fungal genera and Trichoderma compared to VC. This suggests that the addition of T. asperelloides to vermicompost increases the presence of Trichoderma in the samples. 

  1. Some values should add ± SDs, such as Line 229: 2.06 cm, as well as Line 255, 256, 267, 268, 269, 270, 271, 272, 273, 274. 

Response 9: We have added values ± SDs as reviewer’s suggestions.

  1. Figure 2. Trichoderma should be labeled in each figure (B-G).

Response 10: We have labeled with arrows and explain in figure caption as “arrows indicate Trihcoderma colonies.

  1. Line 310. Commonly the higher Shannon value and lower Simpson value represent higher alpha diversity.

Response 11: A higher value of Shannon’s index indicates higher diversity. The Simpson index ranges from 0 to 1, where 0.5 indicates low diversity, 0.5-0.75 indicates moderate diversity, and 0.75-1 indicates high diversity (Maisyroh et al., 2021; Odum, 1971; Simpson, 1949). Based on the results of our study, the TVBC sample showed a Simpson value close to 1, which may represent higher diversity compared to the VC sample.

Reviewer 2 Report

The Authors of this article demonstrated that a vermicompost enriched with the anatgonist fungus Trichoderma asperelloides exerts an antifungal activity against  Rhizoctonia solani, the causal agent of rice blast disease, both in vitro and in planta on rice seedlings. Moreover, they showd that the addition of Trichoderma to the vermicompost enhanced the plant growth promoting effect of this substrate. The activation of vermicompost with Trichoderma induced a disease resistance response in rice seedlings as demonstrated by an increased activity of resistance-related enzymatic activities. The metagenomic analysis revealed the addition of Trichoderma to the vermicompost increased the diversity of mycobiota associated to the rhizosphere of rice seedlings. The subject is of practical interest for agriculture and has been addressed with a rigorous sicentifc approach.

The objectives of the study are clearly indicated. The experimental design and research methods are appropriate. Results are clearly presented and illustrated by graphics. The cited literature, although relatively limited,  is with very few exceptions (see notes in lines 64 and 71), correctly selected and covers all the subjects addressed in the study. The Discussion is consistent with the Results.I introduced only minor text editings and suggestions (see notes in the text, attached PDF file).

Overall the style of the article, including the English style, the experimental design, the presentation of results and the discussion, is concise but, clear and consequential.

The only aspect which was not clear to me was whether the total DNA was extracted from the vermicompost (alone or enriched with Trichoderma ) or from rice roots (see line 198 in M&M). I considered the latter hypothesis. However, if I was wrong, please do not consider the term rhizosphere I added in the Abstract (line 35). In any case make it more clear in the description of M&M (line 198).

Author Response

The Authors of this article demonstrated that a vermicompost enriched with the anatgonist fungus Trichoderma asperelloides exerts an antifungal activity against  Rhizoctonia solani, the causal agent of rice blast disease, both in vitro and in planta on rice seedlings. Moreover, they showd that the addition of Trichoderma to the vermicompost enhanced the plant growth promoting effect of this substrate. The activation of vermicompost with Trichoderma induced a disease resistance response in rice seedlings as demonstrated by an increased activity of resistance-related enzymatic activities. The metagenomic analysis revealed the addition of Trichoderma to the vermicompost increased the diversity of mycobiota associated to the rhizosphere of rice seedlings. The subject is of practical interest for agriculture and has been addressed with a rigorous sicentifc approach.

The objectives of the study are clearly indicated. The experimental design and research methods are appropriate. Results are clearly presented and illustrated by graphics. The cited literature, although relatively limited,  is with very few exceptions (see notes in lines 64 and 71), correctly selected and covers all the subjects addressed in the study. The Discussion is consistent with the Results. I introduced only minor text editings and suggestions (see notes in the text, attached PDF file).

Response 1: Thank you for reviewing this manuscript and providing valuable comments to improve it. We have added more citations and references based on the reviewer's comments in line 64-71.

Overall the style of the article, including the English style, the experimental design, the presentation of results and the discussion, is concise but, clear and consequential.

Response 2: Thank you for your suggestions.

Detail comments

The only aspect which was not clear to me was whether the total DNA was extracted from the vermicompost (alone or enriched with Trichoderma ) or from rice roots (see line 198 in M&M). I considered the latter hypothesis. However, if I was wrong, please do not consider the term rhizosphere I added in the Abstract (line 35). In any case make it more clear in the description of M&M (line 198).

Response 3: DNA extraction was conducted directly on vermicompost. We have omitted the rhizosphere in the abstract and provided more details in the materials and methods section, stating "Total genomic DNA was extracted from a TBVC sample and from VC as a control without plant roots.

Response 4: For more text revision, we used blue font for responding to reviewer 2

Round 2

Reviewer 1 Report

Accept in present form

Accept in present form